# Fine Mapping of a Grain Shape Gene from a Rice Landrace Longliheinuo-Dwarf (*Oryza sativa* L. ssp. *japonica*)

**Fei Shang, Xu Chao, Kaiwen Meng, Xianghe Meng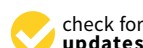, Qin Li, Lifang Chen and Jianfei Wang \***

The State Key Laboratory of Crop Genetics and Germplasm Enhancement, Nanjing Agricultural University, Nanjing 210095, China; 2016201035@njau.edu.cn (F.S.); 2017101122@njau.edu.cn (X.C.); 2014101127@njau.edu.cn (K.M.); mengxianghe_521@163.com (X.M.); qinli0529@163.com (Q.L.); lifangchen_32@163.com (L.C.)

\* Correspondence: wangjf@njau.edu.cn; Tel.: +86-13952000228

**Abstract:** Identification of grain shape genes can facilitate breeding of rice cultivars with optimal grain shape and appearance quality. In this study, we selected two rice germplasms, namely Longliheinuo-dwarf (LH) and N643, with different grain shape, to construct a genetic population for quantitative trait locus (QTL) analysis. A major QTL (*qGS7*), controlling the ratio of grain length to grain width, was mapped on the chromosome 7 in a $BC_1F_4$ line. By high-resolution linkage analysis, *qGS7* was delimited to a 52.8 kb region including eight predicted genes. Through sequence alignment and real-time PCR expression analysis of these ORFs, ORF3 (LOC_Os07g42410) was selected as the candidate gene for further analysis. Single nucleotide polymorphisms (SNP) diversity analysis of ORF3 revealed that a single nucleotide deletion in the 7th exon resulted in a frameshift in parent LH and the parent in which a premature stop codon was identified. It was a rare mutation that caused grain shape difference. Real-time PCR analyses showed that the expression characteristics of ORF3 was in accordance with the development of spikelets. Of the 18 agronomic traits investigation in *qGS7* near isogenic lines (NILs) showed that *qGS7* not only changed grain shape but also affected plant height, panicle curvature, panicle length, the length of second leaf from the top, and chalkiness.

**Keywords:** grain shape; gene mapping; NILs; rice

## 1. Introduction

As one of the most important cereal crops in the world, rice provides staple food for approximately half of the world population [1,2]. Grain shape is a key determinant of grain yield and quality in rice. In general, large grain size is associated with high yield but poor quality [3]. Grains with different shape are preferred by the consumers in different geographical locations worldwide [4]. Therefore, it is of great significance to identify grain shape genes and apply them to rice breeding.

To date, many genes controlling grain shape have been detected and cloned in rice [5]. Multiple signaling pathways are involved in regulating the grain shape, including the ubiquitin–proteasome pathway, protein kinase signaling pathway, phytohormones, transcriptional regulatory factors, and G-protein signaling pathways [6,7]. *GS3* is known to encode a putative trans-membrane protein involved in the G protein signal pathway, which is the first major QTL regulating grain size [8,9]. *GW2* and *GW5* (also known as *qSW5*), negatively regulate rice grain width and weight. *GW2* encodes a RING-type E3 ubiquitin ligase, while *GW5* encodes a calmodulin-binding protein [10–13]. *OsSPL13*, encoded by *GLW7*, acts as an SBP (Squamosa promoter binding protein)-domain transcription factor and positively regulates grain length by affecting cell size [14]. *GW8* encodes the transcription

factor *OsSPL16*, which binds to the *GW7* promoter and regulates grain shape by reducing *GW7* expression [15,16]. *GW6a*, as a histone acetyltransferase, elevates its expression activity and raises grain weight and yield [17]. *qGL3* encodes the protein phosphatase OsPPKL1, which participates in brassinosteroid-signaling in rice, and *GL3.1* controls grain size by regulating cyclin-T1-3 [18–20]. *GS5* and *GS2/GL2* are also involved in the BR signaling pathway [21–23]. *qTGW3* encodes the OsSK41 in the glycogen 38 synthase kinase 3/SHAGGY-like family. OsSK41 interacts with OsARF4 to negatively regulate grain size and weight [24]. *TGW6* and *BG1* also participate in the auxin pathway [25,26]. However, how these signaling pathways regulate the grain shape still awaits further research.

In this study, we used Longliheinuo-dwarf (LH) as the donor parent and N643 as the recurrent parent to develop different mapping populations for grain shape. A major QTL (*qGS7*), controlling the ratio of grain width to grain length, was mapped on the chromosome 7 in a $BC_1F_4$ line. Map-based cloning revealed that *qGS7* was found to be allelic to *EP2/DEP2/SRS1/cl7(t)*. The investigation of agronomic traits in the paired near isogenic lines (NILs) showed that *qGS7* only affected the partially evaluated traits of plants and had no significant effect on the yield per plant. The results not only provide some new information for grain shape but also have an application value in appearance quality breeding in rice.

## 2. Materials and Methods

### 2.1. Plant Materials and Population Development

Longliheinuo-dwarf (LH) was obtained from $Co^{60}$ irradiated Longliheinuo, and it was insensitive to gibberellin. N643 is a semi-dwarf line derived from the cross of two cytoplasmic male sterility maintenance lines, namely Longtepu and Yuefeng.

$BC_1F_2$ plants with various grain shape were selected to develop into $BC_1F_{2\sim4}$ lines. A $BC_1F_4$ line (RHL2401) was found to be separated only in grain shape and used to map the grain-shape gene. Short-grain plants of RHL2401 were back-crossed with N643 to develop a $BC_2F_2$ population for fine mapping. The heterozygotes from the $BC_2F_2$ population were selected to generate $BC_2F_3$ populations for further fine mapping (Figure 1).

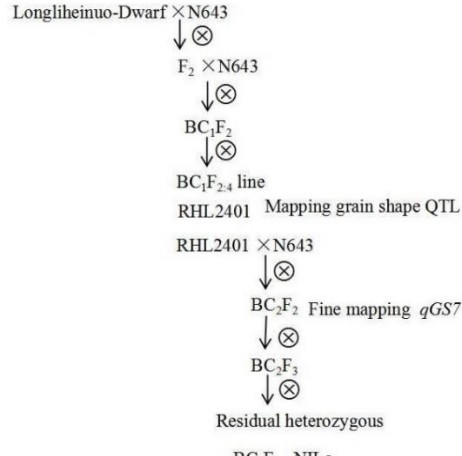

**Figure 1.** Schematic of the generation of the genetic materials used in this study.

In addition, heterozygous plants from the $BC_2$ population were successively self-pollinated and kept residual heterozygous of *qGS7* by marker assisted selection (MAS) to $BC_2F_6$. In the $BC_2F_6$ population, homozygotes were selected to form paired NILs for agronomic trait investigation (Figure 1).

### 2.2. Trait Evaluation

Rice materials were grown in the field at the Experimental Station of Nanjing Agricultural University (Jiangsu Province, China). Field management was carried out in keeping with the local standard methods [27]. Grain length and grain width were measured by electronic vernier caliper (0.01 mm) after the seeds were harvested and dried. Ten fully filled grains were randomly selected from each plant.

### 2.3. Molecular Marker Analysis

Simple sequence repeat (SSR) markers were designed as described on the Gramene database. Insert and deletion (InDel) markers were newly designed by comparison of DNA sequences between Nipponbare and 93-11 at the target region. The genomic DNA of each plant was extracted from fresh rice leaves according to a modified CTAB protocol. PCR was performed as follows: 95 °C for 5 min, followed by 35 cycles of 95 °C for 15 s, 55–60 °C for 20 s, 72 °C for 30 s, and a final elongation step at 72 °C for 10 min. The PCR products were analyzed on 6%–8% nondenaturing polyacrylamide gel.

### 2.4. Genetic Mapping

Bulked segregant analysis (BSA) was applied for the primary mapping. A total of 207 polymorphic SSR markers evenly distributed on 12 chromosomes were used to identify the genotype of individuals. The genetic map construction and QTL analysis were conducted by the composite interval mapping method using the IciMapping 3.1 software. The candidate genes were predicted using the data at http://rice.plantbiology.msu.edu/cgi-bin/gbrowse.

### 2.5. Sequencing Analysis and Expression Analysis

The sequence of gene were downloaded from Rice Genome Annotation Project database. Based on the sequence, primers were designed using Primer Premier 5 (Supplementary Materials Table S1). The genome sequence was amplified from the leaves using Phusion high fidelity DNA polymerase (Takara Biotech, Japan). Three independent purified PCR products were directly sequenced by Springen Biotechnology Co., Ltd. (Nanjing, China). Bio-XM software (Nanjing Agricultural University, China) was used for sequence alignment.

Total RNA was extracted from panicles at four different development stages with the RNA extraction kit (Bioteke, Wuxi, China), including pollen mother cell formation stage (young panicle 2–5 cm), pollen mother cell meiosis (young panicle 5–10 cm), pollen content filling stage (young panicle close to full length), and pollen maturation (anthers turn yellow, spikelets setting). The complementary DNA (cDNA) was synthesized with random oligo nucleotides utilizing a reverse transcription system (Vazyme Biotech, Nanjing, China). Quantitative real-time PCR (20 uL reaction volume) was conducted using 1 uL of cDNA, 0.5 uL of each primer, and 10 uL of AceQ qPCR SYBR Green Master Mix (Vazyme Biotech, Nanjing, China) in a Roche480 real-time PCR detection system. The fold change in cDNA relative to the reference gene (18s-rRNA) was determined by the comparative cycle threshold method. Relative expression levels of gene was calculated using the comparative Ct method [28].

### 2.6. Agronomic Traits Evaluation of Paired NILs for qGS7

Three paired *qGS7* NILs were seeded in nursery trays containing nutrient matrix. After 30 days, the NILs were transplanted to experimental plots in a randomized complete block design with three replications. Each replication had forty plants in a density of 25 cm (4 lines) × 13.3 cm (10 plants). Twenty plants, excluding marginal effects, were collected to evaluated agronomy traits. The agronomy traits were surveyed included plant height, panicles per plant, stem diameter, top internode diameter, flag leaf length, flag leaf angle, length of second leaf from the top, panicle length, panicle exertion length, panicle curvature, primary branch number, secondary branch number, spikelet number per panicle, filled grain number per panicle, grain length, grain width, 1000-grain weight, and chalkiness.

## 3. Results

### *3.1. qGS7 is a Major QTL for Grain Length and Grain Width*

N643 showed slender grains, and LH showed small round grains (Figure 2A). A $BC_1F_4$ line
(RHL2401) was found to have separation in grain shape, and other agronomic traits were approximate
to N643. The frequency distribution of grain length, grain width, and the ratio of grain length to grain
width were both bimodal distributions (Figure 2B–D). The segregation of long-grain to short-grain
plants in RHL2401 closely fitted the ratio of 3:1 ($\chi^2 = 2.41$, $p > 0.05$). The segregation ratio of narrow
grain to wide grain plants also was 3:1 ($\chi^2 = 3.57$, $p > 0.05$). Furthermore, significant negative correlation
($R = -0.776$, $p < 0.01$) between the two traits, suggested that they were controlled by the same gene or
by two closely linked genes.

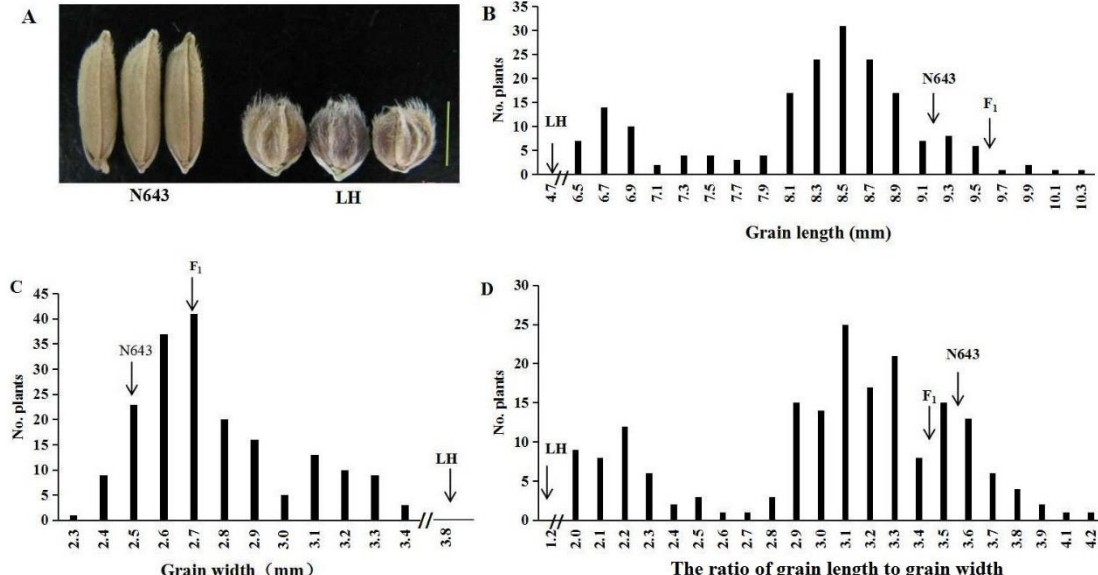

**Figure 2.** Grain shape characteristics of parents and frequency distribution of grain shape in the
RHL2401 population. (**A**) Grain morphology of Longliheinuo-dwarf (LH), N643. Scale bar = 4 mm.
Frequency distribution of grain length (**B**), grain width (**C**), and the ratio of grain length to grain width
(**D**) in the RHL2401 population.

The mixed DNA samples of RHL2401 plants were whole genome genotyped by 207 polymorphic
markers (Supplementary Materials Figure S1). QTL mapping was performed in the heterozygous
intervals with polymorphic markers. Two QTLs for grain length were identified on chromosomes 3
and 7, named *qGL3* and *qGL7*, respectively (Table 1). The *qGL3* was located between SSR markers
RM7370 and RM6832, explaining 7.2% phenotypic variation with a LOD value of 8.85. This interval
had a reported locus *GS3*, which was a major QTL for grain length and weight, and widely presented
in indica rice. *qGL7* was mapped between SSR markers RM6344 and RM8261, explaining 64.55% of
phenotypic variation. It was a major QTL controlling grain length.

Two QTL for grain width were detected on chromosomes 5 and 7, named *qGW5* and *qGW7*
(Table 1). The *qGW5*, between markers RM7444 and RM1089, explained 13.47% phenotypic variation
with LOD value of 15.72, and its additive effect came from LH. In this interval existed a reported
gene *qSW5*. *qGW7* was located between markers RM6344 and RM8261, explaining 60.13% phenotypic
variation with a LOD value of 35.15. Coincidentally, this interval was shared by the *qGL7* and *qGW7*.
This indicated that this fragment of LH was able to increase grain width and inhibit grain length. The
corresponding N643 fragment could increase the grain length and reduce the grain width. A grain
shape QTL for the ratio of grain length to width was also mapped in this interval, which explained
68.67% phenotypic variation with a LOD value of 43.94 (Table 1). It was determined that the single

site, affecting grain length and grain width simultaneously, was named *qGS7* (QTL for grain shape on chromosome 7).

**Table 1.** QTL detected in a $BC_1F_4$ line (RHL2401).

| Traits | Chr | QTL | Flanking Markers | LOD | PVE (%) | Add | Dom |
|---|---|---|---|---|---|---|---|
| Grain Length | 3 | *qGL3* | RM7370-RM6832 | 8.85 | 7.20 | 0.323 | −0.189 |
| | 7 | *qGL7* | RM6344-RM826141.26 | | 64.56 | −0.957 | 0.505 |
| Grain Width | 5 | *qGW5* | RM7444-RM108915.72 | | 13.47 | 0.126 | 0.027 |
| | 7 | *qGW7* | RM6344-RM826135.15 | | 60.136 | 0.268 | −0.133 |
| Grain Length/Grain Width | 7 | *qGS7* | RM6344-RM826143.94 | | 68.67 | −0.606 | 0.268 |

### 3.2. Fine Mapping of qGS7

For fine mapping *qGS7*, new polymorphic InDel markers were developed (Supplementary Materials Table S1). In all, 605 $BC_1F_{2:5}$ plants were genotyped by those markers. Through phenotype assays of populations, *qGS7* was located to an interval between markers In8 and In4 (Figure 3A). To isolate the *qGS7* gene, 2089 plants from the $BC_2F_2$ population were used. The *qGS7* were narrow down to 101 kb region between markers In13 and In4 (Figure 3B). High-resolution mapping was conducted using 6000 $BC_2F_3$ plants to further delimit the *qGS7* locus to a 52.8 kb region that contained eight predicted genes (Figure 3C,D).

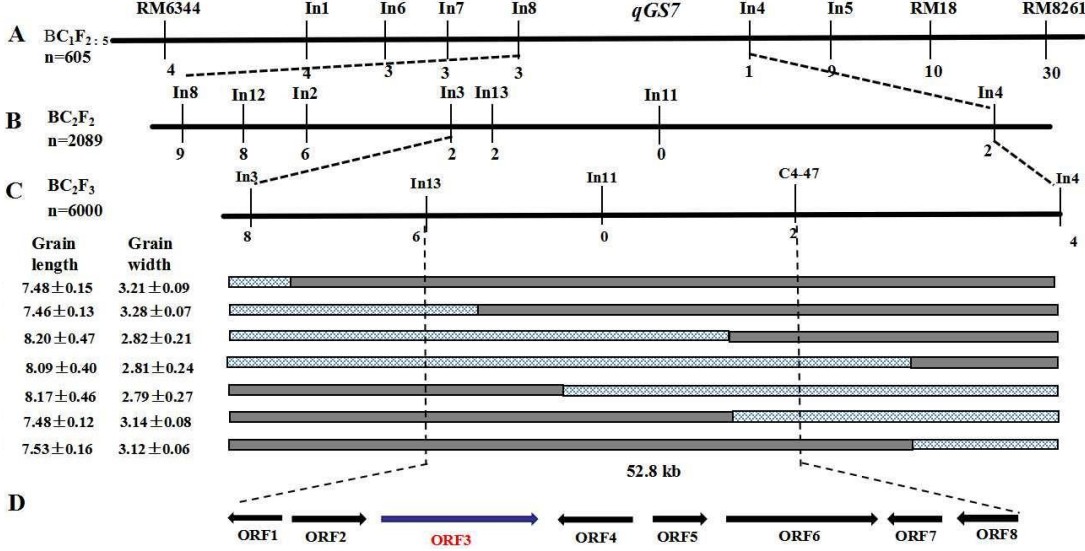

**Figure 3.** Map-based cloning of *qGS7*. (**A**) Physical map of the *qGS7* locus using 605 $BC_1F_{2:5}$ plants. (**B**) Physical map of the *qGS7* locus using 2089 $BC_2F_2$ plants. (**C**) High-resolution linkage analysis of the *qGS7* locus using 6000 $BC_2F_3$ plants. Grey solid bars represent homozygous chromosomal segments for N643 and grille for heterozygote. (**D**) Predicted open reading frames in the target region of the rice genome.

### 3.3. Candidate Genes Predication of qGS7

Among them, ORF2 and ORF8 encoded transposon protein. Therefore, we first cloned and sequenced the remaining six genes. ORF1 had no nucleotide variance between two parents. ORF3 had six SNPs in the coding region between two parents. SNP1 from T (LH) to A (N643) in the 7th exon caused amino acid changes from isoleucine to asparagine. SNP2 from one base deletion of LH resulted in a missense mutation, which changed the 502th and its subsequent amino acid sequence until terminated translation at position 508. SNP3 from T (LH) to G (N643) in the 7th exon caused amino acid changes from leucine to tryptophan. SNP4, SNP5, and SNP6 were synonymous mutations. ORF4 had four SNPs between two parents, but they did not cause amino acid changes. ORF5 had two SNPs between two parents. SNP1 was a synonymous mutation, and SNP2 from A (LH) to T (N643) caused premature termination of translation. ORF6 had one SNP between two parents, which caused amino acid changes from isoleucine to threonine. ORF7 had two SNPs between two parents, of which SNP1 was a synonymous mutation. SNP2 from A (LH) to G (N643) caused amino acid changes from valine to isoleucine (Table 2).

**Table 2.** Candidate genes for the *qGS7* in the 52.8 kb region.

| ORF | Locus Name | Nucleotide Length | Putative Function | Coding Sequence and Amino Acid Change |
|---|---|---|---|---|
| ORF1 | LOC_Os07g42395 | 2636 | DNA-directed RNA polymerase II subunit RPB9 | No difference in coding region |
| ORF2 | LOC_Os07g42400 | 2229 | Transposon protein | - |
| ORF3 | LOC_Os07g42410 | 7850 | Expressed protein | 7 SNPs, SNP1:$T^{1145} \rightarrow A$ caused $Ile^{382} \rightarrow Asn$; SNP2: A base deletion at 1505 caused a frameshift and termination at 1525 position |
| ORF4 | LOC_Os07g42420 | 5661 | 3-oxoacyl-synthase | 4 SNPs, no AA change |
| ORF5 | LOC_Os07g42430 | 1673 | Expressed protein | 2 SNPs, SNP1:$T^{135} \rightarrow C$, no AA change; SNP2:$T^{369} \rightarrow A$ caused a 2 AA deletion at C-terminal |
| ORF6 | LOC_Os07g42440 | 5540 | Glycolate oxidase | 1 SNP, $C566 \rightarrow T$ caused $Ile^{189} \rightarrow Thr$ |
| ORF7 | LOC_Os07g42450 | 2260 | Ribosomal protein S2 | 2 SNPs, SNP1:$C^{488} \rightarrow T$, no AA change SNP2: $A^{772} \rightarrow G$ caused $Val^{258} \rightarrow Ile$ |
| ORF8 | LOC_Os07g42460 | 2268 | Transposon protein | - |

ORF1 and ORF4 did not cause amino acid difference between two parents. Therefore, the expression levels of ORF3, ORF5, ORF6, and ORF7 were examined at four different stages of panicle development in the paired near-isogenic lines (NIL-*qGS7*[N643] and NIL-*qGS7*[LH]). ORF5 and ORF6 showed no expression in the two NILs plants. The expression level of the ORF3 and ORF7 both gradually decreased with panicle development in the NIL-*qGS7*[N643] plants. In the NIL-*qGS7*[LH] plants, they increased at first, then decreased and were barely detectable when the panicles reached their final lengths (Figure 4). Previous studies demonstrated that *ORF3 (EP2/DEP2/SRS1/cl7(t))* was the erect panicle and small round grain gene [29–32]. ORF7 encodes ribosomal protein S2. Consequently, ORF3 (LOC_Os07g42410) was selected as the candidate gene for further analysis.

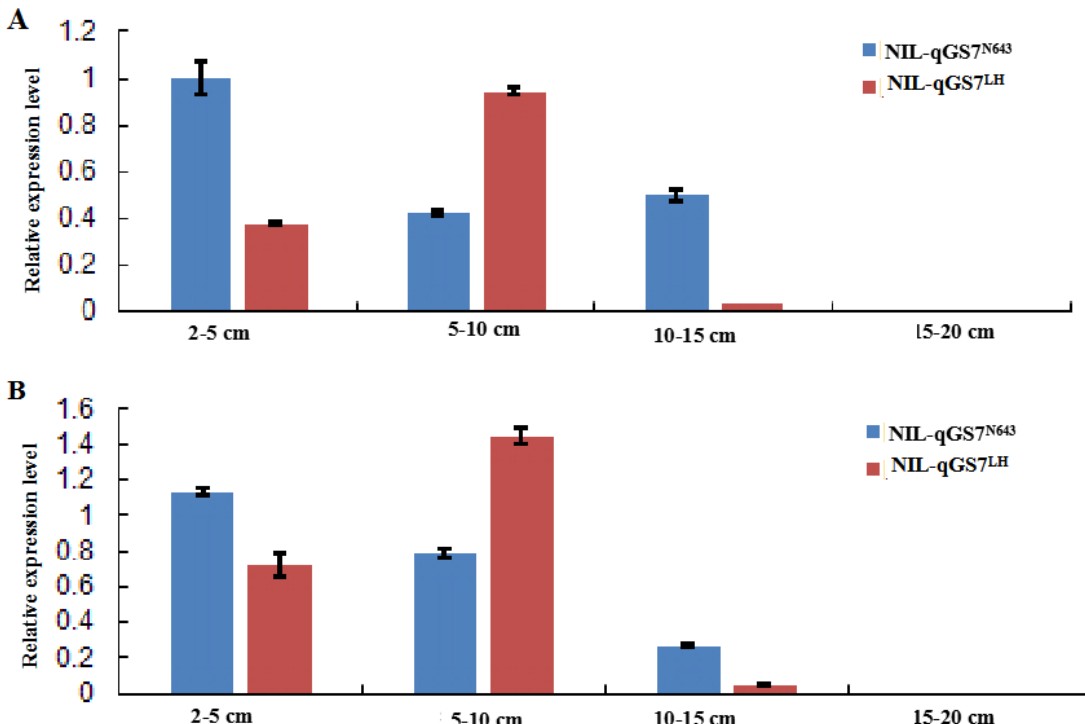

**Figure 4.** Real-time PCR analysis of candidate genes at four different stage with panicle development. (**A**) The expression level of *ORF3*. (**B**) The expression level of *ORF7*. Values are the means ± SD of three replicates.

### 3.4. SNP Diversity Analysis of qGS7

In order to analyze the SNP diversity of *qGS7*, nucleotide sequence analysis of LOC_Os07g42410 was further carried out in 16 rice varieties (including the parents). The sequencing fragments covered three missense mutations in the coding region (Table 3). The long and short grain were distinguished by the ratio of grain length to width. The results showed that variation in SNP1 and SNP3 were not related to grain shape, while SNP2 only occurred in the LH. Therefore, we inferred that SNP2 was the core mutation site that caused the grain shape difference in the two parents and may also be a rare mutation.

**Table 3.** SNP diversity analysis for *qGS7*.

| Accession | Subsp | Grain Length(mm) | Grain Width (mm) | Grain Length/Grain Width | SNP1 | SNP2 | SNP3 |
|---|---|---|---|---|---|---|---|
| | | | | | 1145 | 1505 | 1547 |
| Longliheinuo-dwarf | Japonica | 4.5 | 3.36 | 1.34 | T | - | G |
| N643 | Indica | 8.89 | 2.34 | 3.8 | A | T | T |
| 80-4 | Japonica | 6.96 | 3.28 | 2.12 | T | T | G |
| Jiangdu6102 | Indica | 7.45 | 3.5 | 2.13 | A | T | T |
| J4195/zhen 511 | Japonica | 6.81 | 3.1 | 2.2 | T | T | G |
| Nipponbare | Japonica | 7.41 | 3.29 | 2.25 | T | T | G |
| SWWR | Indica | 6.28 | 2.74 | 2.29 | A | T | T |
| Siyang 87-2566 | Japonica | 7.5 | 3.25 | 2.3 | T | T | G |
| Jiguang1 | Indica | 7.49 | 2.93 | 2.56 | A | T | T |
| 93-11 | Indica | 9.84 | 2.56 | 3.84 | A | T | T |
| IR50 | Indica | 8.99 | 2.29 | 3.92 | A | T | T |
| YR303-304-1-3 | Indica | 9.83 | 2.25 | 4.37 | T | T | G |
| YR3030-12-6-39 | Indica | 9.64 | 2.14 | 4.5 | T | T | G |
| H94-1 | Indica | 9.67 | 2.07 | 4.67 | T | T | G |

### 3.5. Agronomic Traits Evaluation of Paired NILs for qGS7

Of the 18 agronomic traits investigated, there were significant difference in seven agronomic traits between NIL-*qGS7*[N643] and NIL-*qGS7*[LH], including plant height, the length of second leaf from the top, panicle length, panicle curvature, grain length, grain width, and chalkiness. Compared to NIL-*qGS7*[N643], NIL-*qGS7*[LH] showed that the plant height and its second leaves from top were shorter, the panicles length were shorter and erect, and the grains were shorter, wider, and had more chalkiness. These results indicated that *qGS7* had pleiotropic effects. However, there was no significant difference in the yield traits consisting of panicls per plant, filled grain number per panicle, and the 1000-grain weight. These results indicated that *qGS7* affected partial evaluated traits of plants and had no significant effect on yield per plant (Figure 5; Table 4).

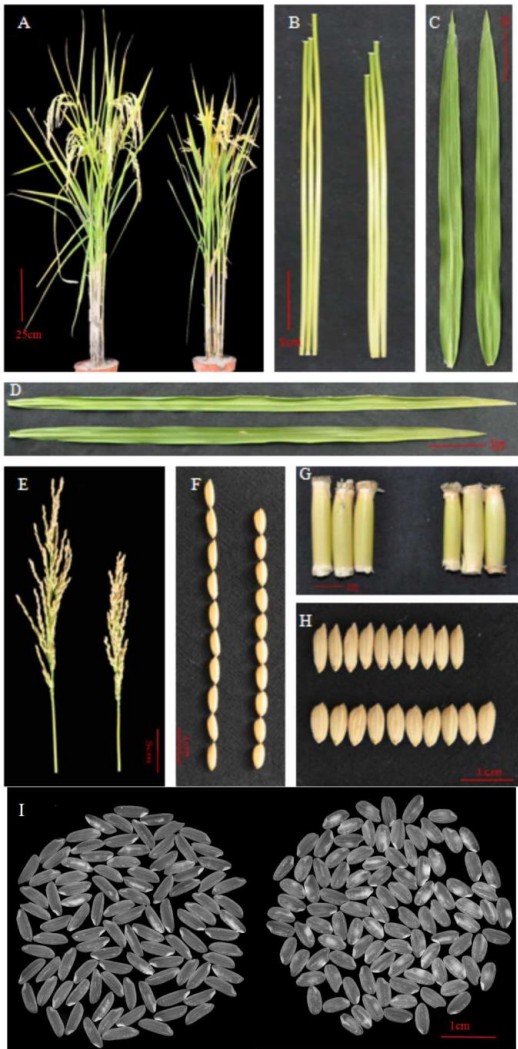

**Figure 5.** Agronomic traits of paired NILs for *qGS7*. (**A**) Gross morphology of *qGS7*[N643] (left) and *qGS7*[LH] (right), bar = 25 cm. (**B**) Top internode diameter *qGS7*[N643] (left) and *qGS7*[LH] (right), bar = 5 cm. (**C**) Flag leave length of *qGS7*[N643] (left) and *qGS7*[LH] (right), bar = 5 cm. (**D**) Second leaf length from the top of *qGS7*[N643] (up) and *qGS7*[LH] (down), bar = 5 cm. (**E**) Main panicle of *qGS7*[N643] (left) and *qGS7*[LH] (right), bar = 5 cm. (**F**) Grain length of *qGS7*[N643] (left) and *qGS7*[LH] (right), bar =1 cm. (**G**) Stem diameter of *qGS7*[N643] (left) and *qGS7*[LH] (right), bar =1 cm. (**H**) Grain width of *qGS7*[N643] (up) and *qGS7*[LH] (down), bar = 1 cm. (**I**) Comparison of the appearance of milled rice between the *qGS7*[N643] (left) and *qGS7*[LH] (right), bar = 1 cm.

**Table 4.** Agronomic traits of paired NILs for *qGS7*.

| Traits | LH | N-643 | NIL-1 | | NIL-2 | | NIL-3 | |
|---|---|---|---|---|---|---|---|---|
| | | | $qGS7^{N643}$ | $qGS7^{LH}$ | $qGS7^{N643}$ | $qGS7^{LH}$ | $qGS7^{N643}$ | $qGS7^{LH}$ |
| Plant Height (cm) | 84.6 ± 1.53 | 89.67 ± 3.06 | 103.43 ± 3.97 | 92.25 ± 3.30 *** | 118.95 ± 3.47 | 105.43 ± 5.38 *** | 115.99 ± 5.12 | 102.95 ± 4.94 *** |
| Panicles Per Plant | 3.00 ± 0.00 | 13.33 ± 2.08 | 10.67 ± 3.93 | 9.70 ± 1.77 | 9.50 ± 1.78 | 10.50 ± 1.96 | 10.10 ± 2.28 | 9.60 ± 1.96 |
| Stem Diameter (mm) | 6.78 ± 0.17 | 5.80 ± 0.20 | 4.94 ± 0.55 | 5.36 ± 0.42 | 5.29 ± 0.79 | 5.05 ± 0.71 | 5.54 ± 0.63 | 5.53 ± 0.48 |
| Top Internode Diameter (mm) | 4.56 ± 0.19 | 3.17 ± 0.17 | 3.24 ± 0.36 | 3.76 ± 0.34 * | 2.86 ± 0.42 | 3.19 ± 0.53 | 3.19 ± 0.49 | 3.46 ± 0.31 |
| Flag Leaf Length (cm) | 22.44 ± 2.33 | 32.00 ± 3.77 | 28.43 ±3.47 | 28.00 ± 9.21 | 28.63 ± 2.03 | 26.95 ± 1.93 | 28.66 ± 2.27 | 27.98 ± 3.02 |
| Flag Leaf Angle (°) | 15.78 ± 0.51 | 11.92 ± 0.14 | 16.07 ± 4.77 | 16.52 ± 3.47 | 25.70 ± 3.58 | 19.43 ± 2.81 | 17.86 ± 2.18 | 21.07 ± 2.57 |
| Length of Second Leaf from the Top (cm) | 36.02 ± 2.72 | 43.25 ± 3.83 | 45.03 ± 3.14 | 39.80 ± 1.75 ** | 46.37 ± 2.23 | 42.42 ± 2.37 *** | 46.39 ± 2.19 | 43.48 ± 2.70 * |
| Panicle Length (cm) | 15.81 ± 0.13 | 19.94 ± 0.93 | 22.69 ± 2.13 | 17.95 ± 3.14 ** | 23.79 ± 1.26 | 18.37 ± 0.91 ** | 23.75 ± 1.34 | 18.86 ± 0.77 *** |
| Panicle Exertion (cm) | 5.58 ± 0.92 | −3.21 ± 1.07 | −1.01 ± 1.70 | −1.56 ± 1.28 | 0.90 ± 0.65 | 2.11 ± 1.06 | 1.32 ± 0.71 | 3.10 ± 1.42 |
| Panicle Curvature (°) | 4.11 ± 0.19 | 104.36 ± 12.39 | 120.96 ± 4.21 | 31.98 ± 12.65 *** | 111.68 ± 9.84 | 60.15 ± 9.99 *** | 113.83 ± 11.89 | 61.93 ± 6.09 *** |
| Primary Branch Number | 13.11 ± 0.38 | 13.65 ± 0.26 | 13.93 ± 1.25 | 14.72 ± 0.74 | 15.18 ± 0.87 | 14.75 ± 0.76 | 15.34 ± 0.88 | 15.26 ± 0.64 |
| Secondary Branch Number | 29.00 ± 5.29 | 30.07 ± 7.42 | 31.02 ± 7.96 | 28.79 ± 3.45 | 34.49 ± 7.80 | 32.47 ± 5.58 | 35.19 ± 3.60 | 36.87 ± 5.18 |
| Spikelet Number Per Panicle | 169.33 ± 18.76 | 171.21 ± 24.00 | 192.21 ± 42.39 | 185.54 ± 15.45 | 209.90 ± 34.32 | 191.75 ± 23.75 | 211.27 ± 17.92 | 211.12 ± 21.83 |
| Filled Grain Number Per Panicle | 131.44 ± 12.22 | 135.49 ± 16.59 | 164.59 ± 36.68 | 160.35 ± 15.54 | 167.40 ± 29.87 | 156.43 ± 19.56 | 179.53 ± 18.86 | 178.10 ± 20.71 |
| Grain Length (mm) | 4.50 ± 0.07 | 8.89 ± 0.04 | 8.00 ± 0.25 | 6.80 ± 0.07 *** | 8.06 ± 0.13 | 7.06 ± 0.11 *** | 8.02 ± 0.14 | 6.97 ± 0.10 *** |
| Grain Width (mm) | 3.56 ± 0.08 | 2.34 ± 0.08 | 2.44 ± 0.05 | 2.86 ± 0.06 *** | 2.50 ± 0.04 | 2.87 ± 0.04 *** | 2.40 ± 0.03 | 2.85 ± 0.03 *** |
| 1000-Grain Weight (g) | 11.77 ± 1.05 | 19.37 ± 0.32 | 18.98 ± 0.47 | 18.75 ± 0.67 | 19.81 ± 0.30 | 19.34 ± 0.44 * | 18.88 ± 0.42 | 18.97 ± 0.39 |
| Chalkiness (%) | glutinous | 10.4 ± 2.88 | 11.25 ± 3.59 | 24.28 ± 6.84 *** | 13.4 ± 2.44 | 34.25 ± 6.23 *** | 15.65 ± 3.65 | 32.30 ± 8.32 *** |

*, ** and *** indicate significant difference at the 0.05, 0.01, and 0.001 probability level, respectively.

## 4. Discussion

Grain size and shape are key agronomic traits and also affect grain yield and grain appearance in rice. Grain width, length, and thickness coordinately determine grain shape [33]. The physical appearance of grain dimensions (length, width, and the ratio of length to width) is essential for grain quality traits in rice. A number of QTLs are known to regulate grain shape in rice, of which several have been cloned and characterized [34]. However, our understanding remains fragmentary, and thus, further efforts to excavate elite genes/alleles are required.

In this study, we selected various grain shape plants of $BC_1F_2$ to develop a QTL mapping population. A major QTL (*qGS7*), controlling the ratio of grain length to grain width, was delimited to a region of 52.8 kb that contained eight predicted genes on chromosome 7 (Figure 3). Sequence alignment of predicted genes showed that the nucleotide variation of ORF3, ORF5, ORF6, and ORF7 caused the change of amino acids. The real time PCR analysis of four genes at different stages of panicle development showed that ORF5 and ORF6 had no expression in the two NILs plants. The expression level of the ORF3 was similar to ORF7 (Figure 4). Previous studies demonstrated that ORF3 (*EP2/DEP2/SRS1/cl7(t)*) was erect panicle and small round grain gene [29–32]. ORF7 encodes ribosomal protein S2. Therefore, we selected ORF3 as the candidate gene for further analysis.

The core mutation site between two parents in ORF3 was identified as SNP2, which a single nucleotide (T) deletion resulted in a frameshift in parent LH and premature termination of the protein. SNP diversity analysis of ORF3 revealed that SNP2 only occurred in LH, and other SNPs were not associated with grain shape (Table 3). We further compared the mutation sites of *cl7 (t)*, *EP2*, *DEP2*, and *SRS1* with *qGS7*. The results showed that although *qGS7* and *EP2/DEP2/SRS1/cl7(t)* belonged to the same gene, they had different mutation site (Supplementary Materials Table S2). The grain size mutation in LH of the *qGS7* candidate gene was novel compared to previous studies [29–32].

Expression levels of *SRS1* were higher in the developing panicle than the fully developed one [29]. *DEP2* gene expression became weaker as the panicles grew longer and was barely detectable when the panicles reached their final lengths [30]. The real time PCR analysis showed that expression level of *qGS7* gradually decreased with panicle development (Figure 4). The result was consistent with previous studies.

Of the 18 agronomic traits investigated in *qGS7* NILs (Figure 5; Table 4), there were significant difference of seven agronomic traits including plant height, second leaf length from the top, panicle length, panicle curvature, grain length, grain width, and chalkiness. However, there was no significant difference in the yield traits. The *qGS7^N643* increased grain length but decreased grain width, resulting in little effect on grain weight and enhanced effect on the ratio of grain length to width. In addition, milled rice from NIL-qGS7^LH showed more chalkiness than that of NIL-qGS7^N643. Although the first internode thickness from the top of NIL-qGS7^LH plants was slightly thicker than NIL-qGS7 ^N643 in this study, they did not show significant differences. The first internode thickness from the top of *EP2/DEP2* mutants is relatively thicker and contains a large number of vascular bundles, which can increase the strength of the internode [30–32].

To date, about 14 QTLs having large effects on grain length and width in rice have been cloned. One of them, *GL7/GW7*, has similar effects on grain length and width with opposite allelic directions, controlling grain shape but hardly influencing grain weight [16,35]. The other 13 genes affect grain size and weight. Four of them mainly control grain width, including *GW2*, *GS5*, *qSW5/GW5*, and *GW8* [11–13,21]. Eight genes mainly control grain length, including *GS2/GL2*, *OsLG3*, *qLGY3/OsLG3b*, *GS3*, *GL3.1/qGL3*, *GL4*, *TGW6*, and *GLW7* [34,36–38]. The remaining one, *GW6a*, has similar effects on grain length and width with the same allelic direction, and exhibits a larger impact on grain weight [17]. *qGS7* and *GL7/GW7*, had similar effects on grain length and width, controlling grain shape without influencing grain weight. This type of QTL could be used for providing a potential gene resource for fine-tuning grain shape to modify grain appearance quality without yield penalty.

**Supplementary Materials:** The following are available online at http://www.mdpi.com/2073-4395/10/3/380/s1, Figure S1. Graphical genotype of RHL2401 line on 12 rice chromosomes; Table S1. The primer used in this study; Table S2. Mutant site of erect panicle and small round gene.

**Author Contributions:** Conceptualization, X.M.; Formal analysis, Q.L. and L.C.; Investigation, X.C., K.M., and F.S.; Writing—original draft, F.S.; Funding acquisition, J.W. All authors have read and agreed to the published version of the manuscript.

**Acknowledgments:** This work was supported by the natural science foundation of Jiangsu Province, China (BK20151427).

**Conflicts of Interest:** The authors have declared that they have no conflict of interest.

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
