# Peer review of "Fine Mapping of a Grain Shape Gene from a Rice Landrace Longliheinuo-Dwarf (Oryza sativa L. ssp. japonica)"

_agronomy, doi:10.3390/agronomy10030380_

Round 1

Reviewer 1 Report

Significant changes were made to the manuscript in the revised version. The manuscript is well written. Mapping the genetic basis of grain size is important for crop improvement. Despite the huge amount of work being invested in this research, identifying the gene underlying grain size QTL in LH as an allele of the previous gene, cl7(t), is a little discouraging.

Major question:

Was allelism test performed to determine the grain size mutation in LH was novel? 

Change Auxine to auxin in the second-page first paragraph.

Reviewer 2 Report

Thank you for addressing the suggestions.

Author Response

Thank you for your comments.

Reviewer 3 Report

Overall, it looks good.

Author Response

Thank you for your positive comments.

Round 2

Reviewer 1 Report

Thanks for incorporating the suggested comments.

This manuscript is a resubmission of an earlier submission. The following is a list of the peer review reports and author responses from that submission.

Round 1

Reviewer 1 Report

The amount of work put into conducting BSA and fine mapping of GS7 is highly commendable. The authors concluded LOC_Os07g42410 as the gene underlying based on previous literature. However, the role of the second gene in grain size was not completely ruled out and this needs to be addressed in the manuscript.

Major question:

How was the second gene ruled out as both genes contain the same expression pattern? Is allelism test done with either DEP2/SRS or cl7(t) to validate GS7 as the gene underlying grain shape? Include these details in the discussion section.

Minor suggestions:

Abstract Line 18 and 19: include the parent in which a premature stop codon was identified. Discussion line 15: references are missing Line 134: Remove ‘QTL detection is performed only in the segregating intervals’. As QTL mapping is done only with polymorphic markers, it is confusing when reading this sentence. Figure 1: Instead of N643 only N64 was included in the parent information Include what gene does ORF7 encode? Include reference ‘Identification and utilization of cleistogamy gene cl7(t) in rice (Oryza sativa L.)’, Journal of Experimental Botany (2014) as cl7(t) is an allele of LOC_Os07g42410.

Reviewer 2 Report

In Figure 4, it is mentioned that you performed three replicates for each sample, however, it is not very clear whether you are referring to biological or technical replicates for the respective panicle stages during the qPCR. It would be nice to clarify and present the data of what might be missing. It is very arbitrary how you selected the different "sizes" of the panicle stages. It would be more useful if the separation was based on developmental parameters instead (e.g. inflorescence, milky stage, grain filling stage, mature grains). In lines 188-191 you claim that you detect 3 SNPs out of which one (SNP2) was responsible for the differences in grain size and shape. However, it is not very clear how did you detect these mutations. For example, was its sequencing of the PCR amplicon itself, was it sequencing of different clones (if yes, then how many clones were submitted per PCR amplicon for sequencing). Moreover, it is not very what kind od Taq polymerase was used for the PCR performance (e.g. normal Taq or Taq polymerase with proofreading activity?). The alignment that shows the mutation in the coding region of LOC_Os07g42410 and is responsible for the phenotypic effects should be presented thein the main manuscript and not as supplementary information. The list of primers that were used in the study should be included. It is not very clear how many plants were used to determine the agronomic parameters presented in Table 2.  

Reviewer 3 Report

Thank you for providing me the opportunity to review this article. 

This article gives an idea of the 18 agronomic traits in qGS7 NILs and the traits that were changed and affected such as grain shape, plant height, panicle curvature, panicle length, etc. 

I really liked the article. I would suggest a minor spelling check or grammar check before the final submit. And, hope that you will continue the work and take it to another level.